# Genetic Evaluation of Body Weights and Egg Production Traits Using a Multi-Trait Animal Model and Selection Index in Thai Native Synthetic Chickens (Kaimook e-san2)

**DOI:** 10.3390/ani12030335

**Published:** 2022-01-29

**Authors:** Kitsadee Chomchuen, Veeraya Tuntiyasawasdikul, Vibuntita Chankitisakul, Wuttigrai Boonkum

**Affiliations:** 1Department of Animal Science, Faculty of Agriculture, Khon Kaen University, Khon Kaen 40002, Thailand; kitsadee_ch@kkumail.com (K.C.); vibuch@kku.ac.th (V.C.); 2Network Center for Animal Breeding and Omics Research, Faculty of Agriculture, Khon Kaen University, Khon Kaen 40002, Thailand; veerayat@kkumail.com

**Keywords:** age at first egg, chicken breast meat, egg number, genetic trend, growth, Thai indigenous chicken

## Abstract

**Simple Summary:**

Native chickens play a vital role in the rural economies of many countries and are considered a valuable genetic resource for use in the development of high-yielding breeds. Native breeds are used as foundation stocks for breeding through a system of crossbreeding with commercial breeds, taking advantage of heterosis. However, due to the constraints of their slow growth rate and low egg production, there is insufficient supply to meet consumer demand. Therefore, improving genetic knowledge is one way to solve these problems and achieve sustainable results. In this paper, a multi-trait animal model and a selection index are developed as a solution for this problem to improve growth and egg performance.

**Abstract:**

To improve the genetics of both growth and egg production, which are limitations in purebred native chickens, new genetic lines can be developed using an appropriate genetic approach. The data used in this study included 2713 body weight (BW0, BW4, BW6, BW8, and BW10), breast circumference (BrC6), chicken age at first egg (AFE), and egg production (240EP, 270EP, 300EP, and 365EP) records covering the period 2015 to 2020. A multi-trait animal model with the average information-restricted maximum likelihood (AI-REML) and a selection index was used to estimate the variance components, genetic parameters, and breeding values. The results showed that males had significantly higher weights than females (*p* < 0.05) from 4 to 10 weeks of age and that this difference increased over the generations. The differences between BW0 and BrC6 by sex and generation were not significant (*p* > 0.05). The estimated heritability of body weight ranged from 0.642 (BW0) to 0.280 (BW10); meanwhile, the estimated heritability of BrC6 was moderate (0.284). For egg production traits, the estimated heritability of 240EP, 270EP, 300EP, and 365EP was 0.427, 0.403, 0.404, and 0.426, respectively, while the estimated heritability of AFE was 0.269. The genetic and phenotypic correlations among the growth traits (BW0 to BW10) were low to highly positive. The genetic and phenotypic correlations between growth (BW0 to BW10) and BrC6 traits were positive, and the genetic correlations between BW6 (0.80), BW8 (0.84), BW10 (0.93), and BrC6 were strongly positive. Genetic correlations among the egg production traits (240EP, 270EP, 300EP, and 365EP) were low to highly positive and ranged from 0.04 to 0.86. The genetic correlations between AFE and all egg production traits were low to moderately negative and ranged from −0.14 to −0.29. The positive genetic correlations between body weight (BW6, BW8, and BW10) and egg production traits were found only in 240EP. The average genetic progress of body weight traits ranged from −0.38 to 30.12 g per generation for BW0 to BW10 (*p* < 0.05); the genetic progress was 0.28 cm per generation for BrC6 (*p* > 0.05). The average genetic progress of cumulative egg production traits ranged from 4.25 to 12.42 eggs per generation for 240EP to 365EP (*p* < 0.05), while the average genetic progress of AFE was −7.12 days per generation (*p* < 0.05). In conclusion, our study suggests that the body weight at six weeks of age (BW6), breast circumference at six weeks of age (BrC6), cumulative egg production at 240 days of age (240EP), and age at first egg (AFE) are the traits that should be used as selection criteria, as they have a positive effect on the development of growth and egg production.

## 1. Introduction

Nowadays, consumers are interested in good-tasting and healthy food enriched with naturally bioactive ingredients. Poultry meat is therefore widely consumed due to its components such as peptide proteins, fatty acids, and antioxidants [1,2,3,4,5], which are found in higher concentrations in native breeds compared with commercial breeds [1,2,3,6,7]. Native chickens are generally raised in almost every household in developing countries including Thailand, which has almost 3 million native chicken farmers with 116,019,532 birds, accounting for 23% of the total poultry production [8]. Most Thai native chickens (TNC) are raised in backyard farms with poor husbandry practices, insufficient nutrition, and hot environments. Genetic limitations in local chickens can lead to low productivity in terms of the growth rate and egg production [1,2,9]. Many studies have demonstrated that the cross breeding of exotic chickens with native chickens results in better performances [10,11]. The synthetic chickens were reported to be compared to purebred native chickens, with a higher growth rate and maintaining meat quality similar to those of native chickens [12,13,14].

In 2010, the Research and Network Center for Animal Breeding and Omics Research, Khon Kaen University, developed a commercial broiler x Thai native (Chee) named Kaimook e-san1 (KM1), which is Thai synthetic chickens (TSC; 50% Thai native genetics). The goals of KM1 breeding are to promote growth performance and egg production compared to TNC. The growth of KM1 was significantly better than that of TNC; TNC was fed until it reached 12 weeks of age, while KM1 was fed until only 10 weeks of age to reach a 1.2 kg market weight [1,2]. The cumulative egg production of TSC was 182 eggs, higher than TNC cumulative egg production of 176 eggs [14]. Additionally, the antioxidant properties of TSC, in terms of anserine and anserine/carnosine, were comparable with those of TNC but higher than in a commercial broiler chicken [3]. However, in our previous studies, we realized that, regardless of whether the native fraction was decreased (i.e., 25% Thai native genetics), the potential bioactive ingredients did not differ from 50% in TSC but were higher than those in commercial broilers [3]. Therefore, it was hypothesized that chickens with 25% Thai native genetics would have better growth performances than those that were 50% Thai native. In 2014, we started to develop the Kaimook e-san2 (KM2: 25% Thai native genetics), which was a cross between the commercial broiler and KM1 (25% Thai native genetics); KM2 has since developed until the sixth generation. KM2 chickens have the following important morphological characteristics: Both males and females are covered with pearly white feathers; the comb type of the male is single, while the females have either single or pea combs; and their shank color is white to yellowish, and the beak color is yellowish in both the males and females. In order to develop and improve the efficiency of the next generation of TSC for market competitiveness, the high productivity of those crossbreeds should be developed first.

Therefore, the objectives of this study were (1) to estimate heritability, genetic correlation, and genetic progress using a multi-trait animal model in TSC (KM2) and (2) to create an appropriate selection index for four primary traits in terms of body weight, breast circumference, age at first egg, and egg accumulation with high genetic values for use as traits in a chicken-breeding program to improve animal genetics in the following generations.

## 2. Materials and Methods

The study was carried out in the experimental farm of the Network Center for Animal Breeding and Omics Research, Faculty of Agriculture, Khon Kaen University, Thailand. The farm is located in the northeastern part of Thailand. The region experiences a hot tropical climate with temperatures ranging from 10 °C in winter to 41 °C in summer. This study was reviewed and approved for institutional animal care based on the Ethics for Animal Experimentation of the National Research Council of Thailand (No. IACUC-KKU-37/64).

### 2.1. Animals, Breeding Plan, and Morphology

A flock of TSC was created by crossbreeding commercial broiler chickens and Thai native chickens called Kaimook e-san1 (50% native chicken). Mixing between the broiler and KM1 was used to create Kaimook e-san2, named KM2 (25% native chicken), in 2014. To date, KM2 has developed until the 6th generation. For the breeding plan, each cock was mated to five hens (artificial insemination) by within-breed selection in a closed nucleus system, and then fertilized to produce eggs that hatched in sets. Three sets of the same parent pairs were bred, each incubated 1 week apart, followed by two more cycles of mating with alternating hens each round; each round produced three sets of offspring. In each generation, there were approximately 1500–1800 chicks, which were selected at 16 weeks of age to produce 30 cocks and 150 hens (effective population size = 100) (called parent stock) to fertilize the next generation. The selection process was divided into two steps: (1) Selection for the desired morphological characteristics of the breed (Figure 1) and (2) selection for individual production traits (the observations more than 1.5 times the interquartile range below the first quartile of each trait were considered outliers and were not used in this analysis).

The morphology shown in Figure 1 was used for the selection of parent stock. Overall, the ideal type traits were strong with a single red comb, a yellow beak, and dark orange eyes. The ear lobes and wattles should be red in color. The ear feathers, hackle feathers, body feathers, saddle feathers, tailfeathers, lesser sickle feathers, and wing feathers should be white pearl in color. The shanks, toes, and claws should be white to yellow in color.

### 2.2. Parent Stock Management

When the chicks were hatched, their weights were recorded, and they were tagged with an ID number. This study was divided into three phases. During Stage 1 (0–4 weeks), called the starter phase, they were raised on a litter floor with brooders using incandescent lighting (24 h of light to 0 h of darkness). During Stage 2 (5–16 weeks), called the grower phase, they were moved to a growing house with a litter floor and raised in a house open to natural light. In the first 2 stages, feed and water were supplied on an ad libitum basis. During stage 3 (16+ weeks), called the mature phase, they were moved to individual battery cages, where lighting (18 h of light to 6 h of darkness) and feed were provided on a limited basis. Males were fed 150 g/bird/day, and females were fed 130 g/bird/day. Mating was accomplished by artificial insemination twice a week and eggs were collected from the hens daily. The chickens were fed a starter ration (19% crude protein, 3% fat, 5% fiber, and 13% moisture) from 0–4 weeks of age, a grower ration (15% crude protein, 2% fat, 6% fiber, and 13% moisture) from 5–16 weeks of age, and a layer ration (17% crude protein, 3% fat, 6% fiber, and 13% moisture) from 16 weeks of age to the end of the production period.

### 2.3. Data Collection

Data were obtained from the Network Center for Animal Breeding and Omics Research, Faculty of Agriculture, Khon Kaen University. A total of 2713 animals with records of body weights, breast circumference, age at first egg, and egg production traits were gathered from six generations during the period of 2015 to 2020. A pedigree file was constructed by tracing back all generations of ancestors and included 3828 individuals born between 2014 and 2019. In the data-editing process, chicks at birth that did not meet the morphology requirements and with a birth weight of less than 26 g were discarded; additionally, we identify outliers for each trait (data more than 1.5 times the interquartile range below the first quartile).

The traits considered included the birth weight (BW0); body weight at 4, 6, 8, and 10 weeks of age (BW4, BW6, BW8, and BW10); and breast circumference at 6 weeks of age (BrC6). The egg production traits included the age at first egg (AFE) and cumulative egg production at 240, 270, 300, and 365 days of age (240EP, 270EP, 300EP, and 365EP).

### 2.4. Genetic Model and Statistical Analysis

The recorded data were validated and analyzed for the least square means, and statistical differences were compared by sex and generation using the post hoc test in the generalized linear model for an unbalanced analysis of variance (GLM procedure) using the SAS package. The variance components and genetic parameters, such as heritability and genetic correlation, were estimated using the average information-restricted maximum likelihood (AI-REML) [15], and the breeding values were estimated using the BLUPF90 Chicken PAK v. 2.5 program [16]. The models used for analysis were as follows:

Multi-trait animal model:Y=Xb+Za+e
where *Y* is the vector corresponding to the phenotypic values for the growth (BW0, BW4, BW6, BW8, BW10, and BrC6) and egg production traits (AFE, 240EP, 270EP, 300EP, and 365EP); *X* and *Z* are incidence matrices related to fixed and random effects, respectively; *b* is the vector of fixed effects, including the chicken hatch set and sex; *a* is the vector of random additive genetic effects, assumed to be $a\sim N\left(0,A\sigma_{a}^{2}\right),$ where *A* is an additive relationship matrix and $\sigma_{a}^{2}$ is the additive genetic variance; and *e* is the vector of random residual effects assumed to be $e\sim N\left(0,I\sigma_{e}^{2}\right),$ where I is the identity matrix and $\sigma_{e}^{2}$ is the residual variance.

### 2.5. Genetic Progress and Selection Index

Genetic progress (∆G)  in each trait was estimated based on the equation $\frac{\Delta G}{gen}=\frac{\sigma_{A}\cdot h\cdot i}{gen}$, where $\sigma_{A}\;$= the standard deviation of additive gene effect, *h* = the accuracy of selection, *i* = the selection intensity (20% per generation), and *gen* = generation (year of birth of chickens). The generalized linear model for an unbalanced analysis of variance (PROC GLM) using the SAS package was used to compare the EBV by generation. The accuracy of the selection index (*r*) by sex was calculated based on the equation $r=\sqrt{\frac{b^{\prime }Gb}{v^{\prime }Gv}}$, where $b=P^{-1}Gv$; *P* = phenotypic variance–covariance matrix; *G* = genetic variance–covariance matrix, and $G=A\sigma_{a}^{2},$ where *A* is an additive relationship matrix and $\sigma_{a}^{2}$ is the additive genetic variance; *v* = the vector of relative economic weights corresponding to the traits considered in this study.

The selection index was calculated based on the estimated breeding value (*EBV*) four traits: Body weight ($EBV_{BW..})$, breast circumference $\;(EBV_{BrC6})$(breast meat is the most nutritious and expensive part), age at first egg ($EBV_{AFE})$(for earlier egg production), and egg production ($EBV_{..EP})$. The relative economic value (ϑ) for each trait was calculated as a proportion of the standardized economic value to the total economic importance of all the traits evaluated in the given production system. We determined that both growth- and egg-related traits were of equal importance and therefore the relative economic values were defined as 0.5 for growth traits and 0.5 for egg traits. When considered in detail, the genetic correlations among growth and breast circumference traits are large and positive, so the relative economic values were given equal proportions of 0.25. Regarding egg-related traits, the cumulative egg production was more economically important than the age at first egg, so the relative economic values were defined as 0.30 and 0.20, respectively. The selection index equation is shown as follows:$$I=\left(\vartheta_{1}\times EBV_{BW..}\right)+\left(\vartheta_{2}\times EBV_{BrC6}\right)+\left(\vartheta_{3}\times EBV_{..EP}\right)-\left(\vartheta_{4}\times EBV_{AFE}\right),$$
where *I* is the selection index; $\vartheta_{1}=0.25,\vartheta_{2}=0.25,\vartheta_{3}=0.30,\vartheta_{4}=0.20\;$ are relative economic values for body weight, breast circumference, egg production, and age at first egg traits, respectively; $and\;EBV_{BW..},EBV_{BrC6},EBV_{..EP},EBV_{AFE}$ are estimates of the breeding values for the traits, which correspond to the economic values.

## 3. Results

### 3.1. Growth and Egg Production Performance

The number of records and descriptive statistics of growth and egg production traits are shown in Table 1. Figure 2 showed the mean body weights for KM2 from hatching to 10 weeks of age and the breast circumference at 6 weeks of age separated by sex and generation. The results showed that males had significantly higher weights than females (*p* < 0.05) from 4 to 10 weeks of age (Figure 2a). At the same time, the birth weight and breast circumference in males and females were not statistically different (*p* > 0.05). The body weight of KM2 tended to increase over the generations, especially in the fifth and sixth generations (Figure 2b). The age at first egg (AFE) and cumulative egg production at 240, 270, 300, and 365 days of age (240EP, 270EP, 300EP, 365EP) are shown in Figure 3. It was found that the mean AFE was 168 days, while the highest and lowest AFE values were 190 and 142 days in the first and sixth generations. Moreover, the mean AFE values for 240EP, 270EP, 300EP, and 365EP were 43, 55, 65, and 78 eggs, respectively.

### 3.2. Estimated Variance Components and Heritability

The estimated variance components and heritability of growth traits are shown in Table 2. The estimated additive genetic variances for body weight ranged from 8.77 g^2^ for BW0 to 12,602 g^2^ for BW10. Residual variances for growth traits ranged from 4.89 g^2^ for BW0 to 32,364 g^2^ for BW10. The estimated heritability of body weight ranged from 0.642 (BW0) to 0.280 (BW10). These values were moderate to high. Meanwhile, the estimated heritability of BrC6 was moderate (0.284).

For egg production traits, the additive genetic variances ranged from 138.10 egg^2^ for 240EP to 489.20 egg^2^ for 365EP. Meanwhile, residual variances ranged from 185.40 egg^2^ for 240EP to 660.40 egg^2^ for 365EP. The estimated heritability of egg production ranged from 0.427 (240EP) to 0.426 (365EP). These values were high. Meanwhile, the estimated heritability of AFE was moderate (0.269).

### 3.3. Genetic and Phenotypic Correlation Estimates

Genetic and phenotypic correlations among growth, breast circumstance, and egg production traits are shown in Table 3. The genetic correlations between body weight (BW0, BW4, BW6, BW8, and BW10) and breast circumference (BrC6) traits were mildly to strongly positive and ranged from 0.08 to 0.93. In contrast, the genetic correlations between body weight (BW0, BW4, BW6, BW8, and BW10) and egg production traits (240EP, 270EP, 300EP, and 365EP) were negative to positive and ranged from −0.48 to 0.15. Positive genetic correlations between body weight (BW6, BW8, and BW10) and egg production traits were found only in 240EP.

Genetic correlations among the egg production traits (240EP, 270EP, 300EP, and 365EP) were mildly to strongly positive and ranged from 0.04 to 0.86. The genetic correlations between AFE and all egg production traits were mildly to moderately negative and ranged from −0.14 to −0.29. Similarly, the genetic correlations between AFE and body weight (BW0, BW4, BW6, BW8, and BW10) and BrC6 traits were negative and ranged from −0.03 to −0.27. The phenotypic correlations between growth and egg production traits were slightly lower than the genetic correlations.

### 3.4. Genetic Progress (ΔG/Generation)

The genetic progress per generation for (a) body weight and breast circumference and (b) age at first egg and egg production traits are shown in Figure 4. The average genetic progress of body weight traits ranged from −0.38 to 30.12 g per generation for BW0 to BW10, tending to increase linearly from BW0 to BW8 and thereafter decreasing for BW10 (*p* < 0.05). The genetic progress of the BrC6 trait was 0.33 cm per generation (*p* > 0.05) (Figure 4a). The average genetic progress of cumulative egg production traits ranged from 4.25 to 12.42 eggs per generation for 240EP to 365EP (*p* < 0.05), while the average genetic progress of AFE was −7.12 days per generation (*p* < 0.05) (Figure 4b).

### 3.5. Selection Index

The top 20% of the selection index values in male and female KM2 chickens are shown in Figure 5. The estimated breeding values (EBV) were used in the calculation of the selection index values. The results showed that the top 20% males (12.98) had a higher selection index than the top 20% females (10.45). The accuracy of the selection index was 3.86% greater for the male chickens than for the female chickens.

## 4. Discussion

To develop high growth and egg productivity for TSC, the crossbreeding of exotic and native chickens with genetic evaluation is needed for genetic selection. This study demonstrated that performing genetic selection during six generations caused an increase in body weight and egg production. The estimates of the heritability and genetic correlations between traits suggest that body weight at 6 weeks of age (BW6), breast circumference at 6 weeks of age (BrC6), cumulative egg production at 240 days of age (240EP), and age at first egg (AFE) are desirable traits to use in a selection index, as selection for these traits has a positive effect on growth and egg production. Finally, a multi-trait animal model and selection index can be used in this population of native chickens to improve growth and egg performance.

The mean body weights of KM2 among generations demonstrated that the selection method used in our study was effective, as the body weights increased over the generations (Figure 2b). Compared with previous reports on TNC and TSC breeds, all ages had a higher mean than TSC [17], TNC [18,19], the India crossbred chicken [11], and the Horro chicken of Ethiopia [20]. Selection for rapid early growth to market weight (1.2 kg) is the most common method of selection in native chickens. The growth of KM1 was significantly better than that of TNC; TNC was fed until 12 weeks of age, while KM1 was fed until only 10 weeks of age to reach a 1.2 kg market weight [1,2]. Our results showed that KM2 chickens had reached market weight by the age of 8 weeks. This is earlier than their parent stock (KM1). It is inferred that the growth performance of KM2 has been greatly improved by genetic selection, thereby resulting in a reduced production cost.

For egg production traits, the mean age at first egg was lower (AFE) and the average cumulative egg production increased (240EP, 270EP, 300EP, and 365EP) over the generations, especially from the fourth generation onwards (Figure 3). Both the age at first egg and egg production traits were higher than in previous reports [21,22,23]. Interestingly, this result demonstrated the importance of AFE, which was related to egg production. However, we could infer that the selection methods used in the present study are effective based on two criteria: The phenotype of the chicken breed and the production performance, with the top 20% of the chicken flock used as the parent stock for the next generation as the results of growth, breast circumference, and egg production have been greatly improved over generations as demonstrated in Figure 2 and Figure 3.

As shown in Figure 3, egg production increased from the fourth generation onward. Even though we could not determine the reason for this change, as AFE decreased, egg production increased. The hens with a lower age at first egg had higher egg production, because these traits are negatively genetically correlated. It has been suggested that delayed puberty has a negative effect on the female reproduction system, especially on ovarian function [24]. Moreover, the late-maturing hens had a shorter laying cycle, which explains why the egg production of the delayed-puberty groups (the first to third generations) was worse. Additionally, the selection intensity in this study increased from the fourth generation by increasing the number of chickens in the replacement flock.

The heritability estimates for the traits in the present study were medium to high (ranging from 0.269 to 0.642; see Table 2), similar to the results of studies carried out in local Venda chickens [25], Mazandaran native chickens [22], and Thai native chickens [18,19,21]. This demonstrated that genetics influenced these traits, which are sufficient for genetic evaluation with acceptable accuracy. The highest heritability for body weight was exhibited at 6 weeks of age (not including day 0 chicks), then tended to decrease with increasing age. Similar results were observed by Dana et al. [20], Saatchi et al. [26], and Manjula et al. [27]. The high heritability for body weight at day 0 is due to the inclusion of the maternal genetic effect [21,22,23]. For egg production traits, the high heritability for cumulative egg production did not differ from 240EP to 365EP (ranging from 0.427–0.426), which was similar to the results of studies carried out in Iranian native fowl [28] and Nigerian local chicken [23] but higher than those found by studies conducted in the Horro chicken of Ethiopia, the Mazandaran native chicken, and Korean native chickens [20,22,29]. Therefore, we suggest that selection for growth and egg production should be performed at 6 weeks of age and 240EP, respectively, thereby reducing the generation interval and cost.

The estimates of genetic and phenotypic correlations between the observed traits varied from low to high (Table 3). The genetic correlation between BW0 and other traits was low, similar to studies conducted in KM1 [17], Esfahan native chickens [30], and Mazandaran native chickens [31]. This indicates that, even with a high heritability for body weight at day 0, BW0 may not be used to accurately predict whether a chicken has the genetic potential for healthy growth at increasing ages. On the other hand, the genetic correlations among body weights at other ages were positive with moderate to high levels, similar to the results of studies conducted in the Horro chicken of Ethiopia [20], the Vanaraja male line chicken [11], and the Thai crossbred black-bone chicken [32]. When considering the correlation, we suggest that selection for growth to 6 weeks of age could be a good trait for a selection program, as it was highly correlated with BW8, BW10, and BrC6 (0.91, 0.79, and 0.80, respectively). The genetic correlation was slightly positive (0.08) between BW6 and 240EP traits. Therefore, simultaneous selection for high body weight and high egg production traits could potentially improve both traits. This finding is interesting because the growth production traits generally showed a negative correlation with egg production traits or no correlation between the two traits [22,33,34]. However, there are some reports showing a positive correlation between growth and egg production traits, such as Dana et al. [20], who showed that the genetic correlation between body weight at 16 weeks of age and the cumulative number of eggs produced from months 1 to 2 (EP12), 3 to 6 (EP36), and 1 to 6 (EP16) had a strong positive correlation (0.69–0.92) in the Horro chicken of Ethiopia. One reason for this is the pleiotropic genes associated with both body weight and egg production. For example, the growth hormone gene, in addition to being associated with an increase in body weight, can also result in an increase in egg production [35,36,37].

The study of genetic trends is a way to monitor the selection process. According to the genetic progress for body weight and egg production shown in Figure 4, selection for these traits was effective. Moreover, it could be confirmed that the growth and egg production traits could be selected simultaneously. Regarding the accuracy of the selection index (Figure 5), we found that the accuracy value of males was higher than that of females (0.781 and 0.752, respectively). This might be a result of the number of male chickens being greater than the number of female chickens. Factors contributing to higher accuracy include (1) the amount of data and the data connectedness, i.e., if less data are available and the data are lowly correlated, the accuracy will be low [38,39]; (2) the use of data from multiple sources, especially individual measurements on each animal, in combination with measurements on relatives and progeny, will provide higher accuracy than using data from a single source [40]; and (3) traits with high heritability result in higher accuracy values as the accuracy is the square root of heritability [41]. Therefore, we inferred that a multi-trait animal model and a selection index approach could be used for accurate genetic selection in both male and female chickens.

## 5. Conclusions

We propose the use of a multi-trait animal model and selection index in this population of native chickens to improve growth and egg performance. Moreover, we determined that the method based on estimating the breeding value is the optimal method for genetic improvement. Body weight at 6 weeks of age, breast circumference at 6 weeks of age, cumulative egg production at 240 days of age, and age at first egg are the traits that should be used in a selection index.

## Figures and Tables

**Figure 1 animals-12-00335-f001:**
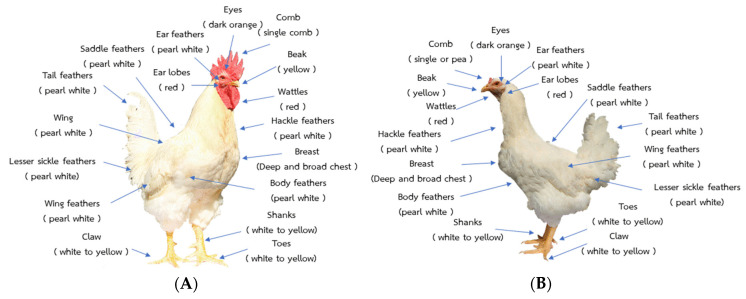
Morphology of male (**A**) and female (**B**) KM2 chickens.

**Figure 2 animals-12-00335-f002:**
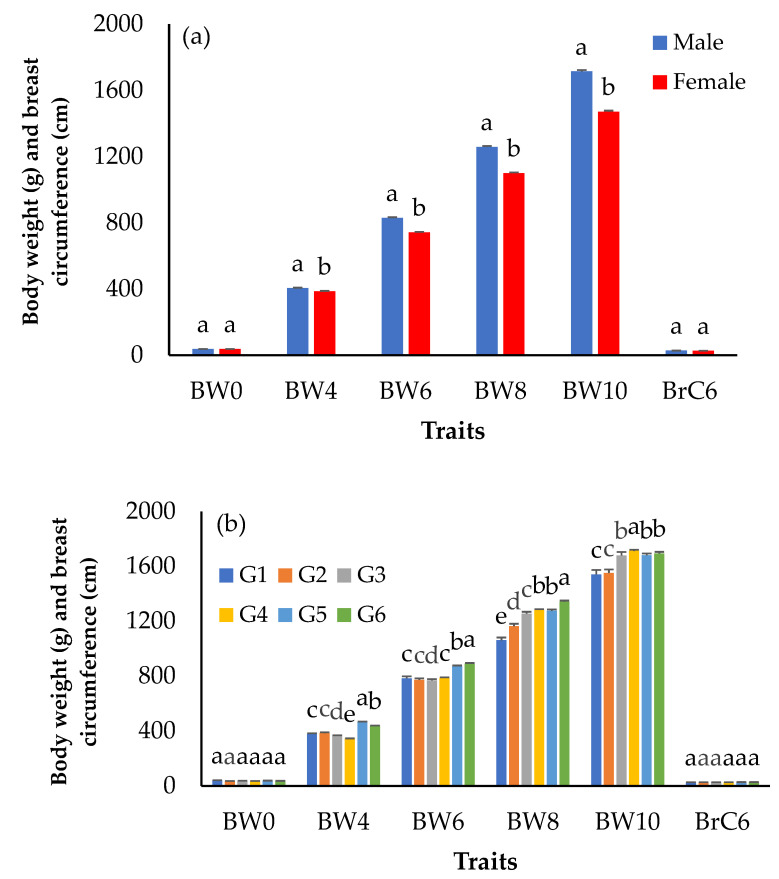
Least-squares means of body weight and breast circumference traits separated by (**a**) sex and (**b**) generation in KM2 chickens; a, b, c, d, e: Means for the trait with different letters differ significantly at *p* < 0.05. BW0—birth weight; BW4—body weight at 4 weeks of age; BW6—body weight at 6 weeks of age; BW8—body weight at 8 weeks of age; BW10—body weight at 10 weeks of age; BrC6—breast circumference at 6 weeks of age; G—generation.

**Figure 3 animals-12-00335-f003:**
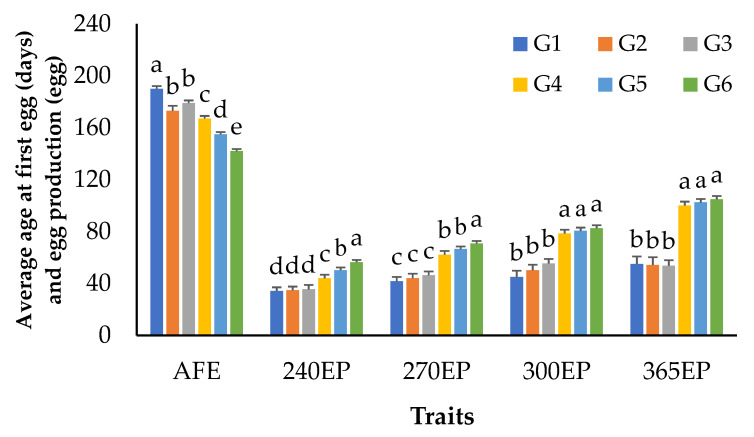
Least-squares means of age at first egg and cumulative egg production traits separated by generation in KM2 chickens; a, b, c, d, e: Means for the same trait with different letters differ significantly at AFE—age at first egg; 240EP—cumulative egg production at 240 days of age; 270EP—cumulative egg production at 270 days of age; 300EP—cumulative egg production at 300 days of age; 365EP—cumulative egg production at 365 days of age; G1-G6—1st generation to 6th generation of chicken.

**Figure 4 animals-12-00335-f004:**
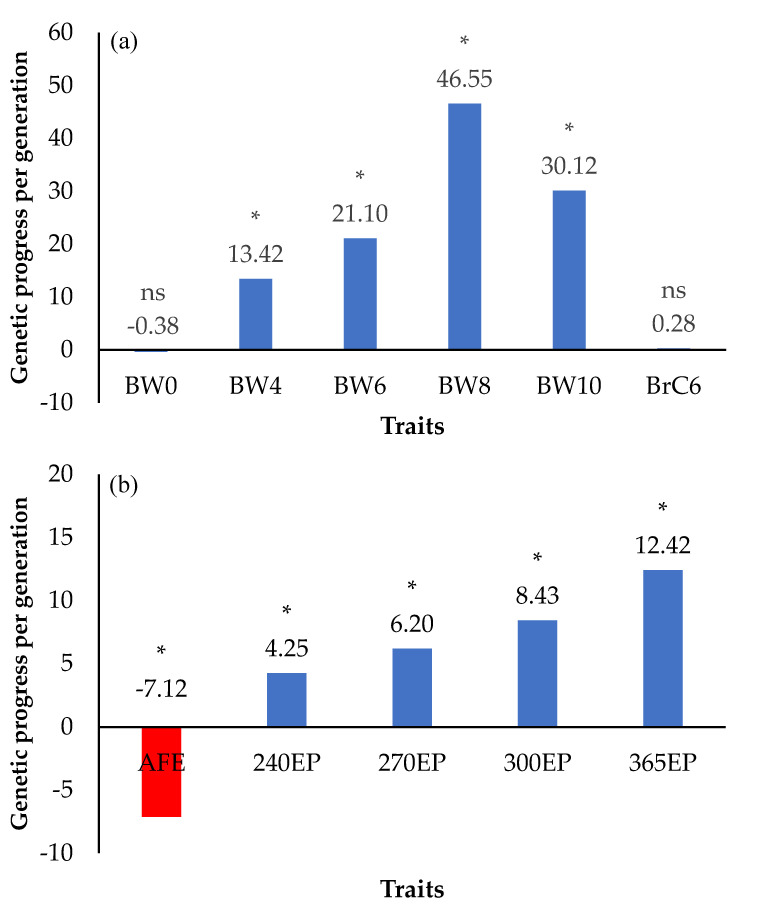
Genetic progress per generation for (**a**) body weight and breast circumference and (**b**) age at first egg and cumulative egg production traits in KM2 chickens. ns is not significant difference (*p* > 0.05); * is significant value within trait by generation (*p* < 0.05). BW0—birth weight (g); BW4—body weight at 4 weeks of age (g); BW6—body weight at 6 weeks of age (g); BW8—body weight at 8 weeks of age (g); BW10—body weight at 10 weeks of age (g); BrC6—breast circumference at 6 weeks of age (cm); AFE—age at first egg (day); 240EP—cumulative egg production at 240 days of age (egg); 270EP—cumulative egg production at 270 days of age (egg); 300EP—cumulative egg production at 300 days of age (egg); 365EP—cumulative egg production at 365 days of age (egg); The blue bar represents a positive numbers for the genetic progress for each trait; The red bar represents a positive numbers for the genetic progress for each trait.

**Figure 5 animals-12-00335-f005:**
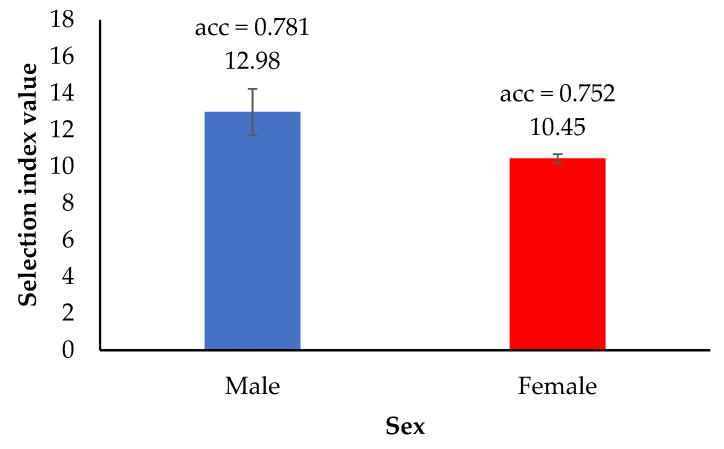
Top 20% of the selection index values and accuracy of the selection index values in male and female KM2 chickens; acc—the accuracy of selection index; The blue bar represents a selection index value for the male; The red bar represents a selection index value for the female.

**Table 1 animals-12-00335-t001:** Descriptive statistics of growth and egg production traits for Kaimook e-san2 chickens.

Traits	No. of Records	Mean	SD	Min	Max	CV (%)
BW0 (g)	2713	37.9	4.2	26.0	50.0	11.1
BW4 (g)	2713	395.4	104.7	150.0	650.0	26.5
BW6 (g)	2713	812.3	138.7	457.0	1400.0	17.1
BW8 (g)	2713	1228.8	220.2	700.0	1895.0	17.9
BW10 (g)	2713	1642.4	261.4	900.0	2450.0	15.9
BrC6 (cm)	2713	27.6	2.3	20.3	36.0	8.3
AFE (day)	2713	168.0	21.0	120.0	199.0	12.5
240EP (egg)	2710	43.0	19.4	12.0	96.0	45.1
270EP (egg)	2710	55.0	24.2	15.0	123.0	44.0
300EP (egg)	2708	65.0	28.1	23.0	146.0	43.2
365EP (egg)	2700	84.0	28.0	44.0	151.0	33.3

No. of records—number of data records; Mean—average of growth and egg production traits; SD—standard deviation; Min—minimum values; Max—maximum values; CV—coefficient of variation; BW0—birth weight; BW4—body weight at 4 weeks of age; BW6—body weight at 6 weeks of age; BW8—body weight at 8 weeks of age; BW10—body weight at 10 weeks of age; BrC6—breast circumference at 6 weeks of age; AFE—age at first egg; 240EP—cumulative egg production at 240 days of age; 270EP—cumulative egg production at 270 days of age; 300EP—cumulative egg production at 300 days of age; 365EP—cumulative egg production at 365 days of age.

**Table 2 animals-12-00335-t002:** Estimated variance components and heritability (standard error) of body weight, breast circumference, age at first egg, and egg production traits in KM2 chickens.

Traits/Parameters	$\sigma_{a}^{2}$	$\sigma_{e}^{2}\;$	$\sigma_{T}^{2}\;$	$h^{2}$
BW0	8.77	4.89	13.66	0.642 (0.10)
BW4	1412	2835	4247	0.332 (0.05)
BW6	5782	7687	13,469	0.429 (0.04)
BW8	11,485	21,059	32,544	0.353 (0.04)
BW10	12,602	32,364	44,966	0.280 (0.02)
BrC6	1.10	2.77	3.87	0.284 (0.03)
AFE	86.50	234.80	321.30	0.269 (0.03)
240EP	138.10	185.40	323.50	0.427 (0.03)
270EP	2.50	330.20	552.70	0.403 (0.02)
300EP	303.40	448.20	751.60	0.404 (0.02)
365EP	489.20	660.40	1149.60	0.426 (0.02)

$\sigma_{a}^{2}$—additive variance; $\sigma_{e}^{2}$—residual variance; $\sigma_{T}^{2}$—total variance; $h^{2}$—heritability; BW0—birth weight; BW4—body weight at 4 weeks of age; BW6—body weight at 6 weeks of age; BW8—body weight at 8 weeks of age; BW10—body weight at 10 weeks of age; BrC6—breast circumference at 6 weeks of age; AFE—age at first egg; 240EP—cumulative egg production at 240 days of age; 270EP—cumulative egg production at 270 days of age; 300EP—cumulative egg production at 300 days of age; 365EP—cumulative egg production at 365 days of age.

**Table 3 animals-12-00335-t003:** Genetic correlations (above the diagonal) and phenotypic correlations (below the diagonal) between body weight, breast circumference, age at first egg, and egg production traits in KM2 chickens.

Traits	BW0	BW4	BW6	BW8	BW10	BrC6	AFE	240EP	270EP	300EP	365EP
BW0	-	0.22	0.15	0.10	0.08	0.12	−0.03	−0.15	−0.03	−0.28	−0.29
BW4	0.16	-	0.61 *	0.51 *	0.40 *	0.30 *	−0.10 *	−0.24 *	−0.32 *	−0.43 *	−0.48 *
BW6	0.14	0.58 *	-	0.91 *	0.79 *	0.80 *	−0.18 *	0.08 *	−0.46 *	−0.49 *	−0.40 *
BW8	0.10	0.48 *	0.85 *	-	0.86 *	0.84 *	−0.20 *	0.14 *	−0.40 *	−0.44 *	−0.32 *
BW10	0.06	0.36 *	0.72 *	0.82 *	-	0.93 *	−0.25 *	0.15 *	−0.38 *	−0.42 *	−0.35 *
BrC6	0.12	0.24 *	0.72 *	0.80 *	0.90 *	-	−0.27 *	0.08 *	−0.32 *	−0.35 *	−0.30 *
AFE	0.01	−0.08 *	−0.16 *	−0.21 *	−0.22 *	−0.24 *	-	−0.14 *	−0.29 *	−0.21 *	−0.16 *
240EP	−0.08	−0.14 *	−0.01 *	0.10 *	0.12 *	0.07 *	−0.14 *	-	0.34 *	0.22 *	0.04 *
270EP	−0.11	−0.25 *	−0.40 *	−0.38 *	−0.36 *	−0.33 *	−0.23 *	0.33 *	-	0.60 *	0.40 *
300EP	−0.20	−0.40 *	−0.42 *	−0.40 *	−0.40 *	−0.28 *	−0.18 *	0.19 *	0.54 *	-	0.86 *
365EP	−0.24	−0.42 *	−0.38 *	−0.28 *	−0.32 *	−0.26 *	−0.12 *	0.09 *	0.38 *	0.85 *	-

BW0— birth weight; BW4—body weight at 4 weeks of age; BW6—body weight at 6 weeks of age; BW8—body weight at 8 weeks of age; BW10—body weight at 10 weeks of age; BrC6—breast circumference at 6 weeks of age; AFE—age at first egg; 240EP—cumulative egg production at 240 days of age; 270EP—cumulative egg production at 270 days of age; 300EP—cumulative egg production at 300 days of age; 365EP—cumulative egg production at 365 days of age; * is a significant value (*p* < 0.05).

## Data Availability

The data presented in this study are available on request from the Network Center for Animal Breeding and Omics Research, Faculty of Agriculture, Khon Kaen University, Thailand.

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
