# Peer review of "Genetic Evaluation of Body Weights and Egg Production Traits Using a Multi-Trait Animal Model and Selection Index in Thai Native Synthetic Chickens (Kaimook e-san2)"

_animals, 2022, doi:10.3390/ani12030335_

Round 1

Reviewer 1 Report

The paper presents is meaning for native chicken breeding. It is a topic of interest to the researchers in the related areas but the paper needs significant improvement before acceptance for publication. My detailed comments are as follows:
1. It is noted that your manuscript needs carefully editing by someone with expertise in technical English editing paying particular attention to English grammer, spelling, and sentence strcture so that the goals and results of the study are clear to the reader. And the acdemic words should be appeared not the usual words: like line103-104.
2.I think the the vital and detailed data of body weitht and egg number should be included, and the methods should be refered in the abstract.
3.In the intrduction, there should be supplied the detailed data about morphology shown in Figure 1 used for the selection of parent stock in line95, not only the figures; there was multiple traits you refered, but which was the important ones? because we are hard to consider all traits; how many chickens was reared in each generation? if the flock is not enough large, the results of your heretility will be no meanings.
4.The materails and methods in your studies was not clear enough. The information of parent like age, body weight and egg production should specified in methods. The methods of different traits collection should be decleared; otherwise, I am interested in the daily food: (1) which kind of boriler chikcen was used to produce TSC chicken in line84? (2)how and why do you maintain the same food for each chicken everyday, and what is the meanings of your design? (3)different food for different sex maybe another factor which would influence the body weight for male and female. 
5.Please check your data in table and figures carefully: like the average daily gain in line164, I can not find the data in any table and figure.
6. The contents in line207-line209, the genetic correlation data describtion should also signified the vital points, and the most important clues which is benifit for your colclusion. Furthermore,  it was hard to understand the data in line 208 and 209, please state your data clearly; and the formula in line248 should appear in the method section.
7. In the discussion, you compared the KM2 and native chickens published by others. can you have the body weight and egg production of native chicken in different KM2 generation? 

Reviewer 2 Report

COMMENTS

This study was conducted for genetic evaluation of body weights and egg production traits using a multivariate mixed model and selection index in Thai native crossbred chickens. I read this paper with interests. However, there are some issues to be addressed before making any decision of the publication in Animals.

[Major comments]

  1. P7L221: In Table 2, the authors should present the phenotypic correlation coefficients together with the current genetic correlation coefficients. If there are obvious difference between the two coefficients, please discuss possible reasons of the discrepancy.

  1. P7L221: In Table 2, p-value levels of each correlation coefficients in Table 2 should be presented.

  1. P10L312: Please explain possible biological regions of positive genetic correlation between 240EP and BW6.

  1. P6L197: Please explain why 0.42(0.05) and 0.45(0.04) are presented in bold numbers.

  1. P8L239: In Figure 4, * is significant different from what?

  1. P4L133: Please add the assumption of the mean and distribution of ?? and ??.

  1. P9L248: The authors should add the description concerning how the relative economic values (?1, ?2, ?3, ?4) were determined.

[Miner comment]

  1. P7L218 should be moved to discussion section or could possibly be deleted.

Reviewer 3 Report

Manuscript ID: animals-1488925

Title: Genetic Evaluation of Body Weights and Egg Production Traits 2 using a Multi-Trait Animal Model and Selection Index in Thai 3 Native Crossbred Chickens (Kaimook e-san2)

Recommendation: Major Revisions
In 1991, M. Georges stated that within a few years BLUP should not be necessary in animal breeding due to the emergence of molecular genetics methods. After 30 years, BLUP (with various modifications, e.g. GBLUP) is still the basic method for estimating the breeding value of animals. For small local poultry populations the classical BLUP method seems to be the only one. Moreover, many methods used in poultry breeding are protected by secrecy by global breeding corporations. Therefore, this, as well as any other publication dealing with the estimating of breeding value of poultry, is a valuable source of knowledge. Unfortunately, there are many mistakes in this article that's need to be corrected.

Comments: Would it not be better to call these birds: Thai native synthetic chickens or Thai native synthetic line. Crossbreeding can erroneously indicate F1, F2 generations as a result of matings between breeding lines.

Lines 11-13: whether other (commercial) breeds do not have such qualities (essential substances, antioxidants, and anti-aging properties) - sentence needs to be reworded because it may be misleading.

Lines 16-17:  Sentence needs to be reworded.  “A multi-trait animal model and selection index, used in the present study, is a new finding”  - Is it really a new finding, it is more appropriate in this context to use: a new solution for this population. Moreover, the method of estimating breeding value alone is not the way to genetic progress. The way can be selection on the basis of breeding value which is estimated by this model.

Part 2.1. Animals, breeding plan, and morphology: There is no information on when the Kaimook e-san2 line was developed. How long has it been an established population (without adding new genes). Please complete this information.

Lines 121-123: For traits (240EP, 270EP, 300EP, 365EP), the authors use the term “days of maturity”. Shouldn't it be "days of life" or “days of age”. If so, please correct. 

Lines 123-124: The number "2713" indicates that approximately 540 birds were evaluated per year, while line 91 states that 1500-1800 birds per generation were selected. Please explain.

Lines 126-128: Which post - hoc test was used - please indicate in the text.

Lines 160-162: This sentence is unnecessary - it gives no information.

Lines 162-163: I understand that the authors in this sentence wanted to say that males had significantly higher weight - please correct.

Lines 163-165: In the sentence is  "average weight" and it should be: “average weight gain” - please correct.

Figures: All abbreviations used in figures should be explained (as in tables), either on the figure box or in the title of the figure. 

Figure 2: The probability on the graph box is unnecessary.

Figure 1 and 2: In titles instead "p < 0.05 is considered a statistically significant difference" should be: a, b, c, d - bars for the trait with different letters differ significantly at P?<?0.05 (?????? test).

Figure 3: From the fourth generation, the laying rate increases rapidly - what is the reason for this change, please explain.

Table 1: In Table 1, for the BW4 to BW10 variances, integers are sufficient, and for the AFE and all EP variances, one place after the dot is sufficient.  

Lines 218-219: Why? - please explain.

Table 2: Why the selected correlations are bolded - please explain in the table title.

Figure 4, Title: should be “ns is not significant value (p > 0.05); * is significant value (p < 0.05).”

Lines 26-30, lines 255-259, lines 320-324: I don't understand the authors' intent. The selection uses a four-trait index and the authors suggest selection based on BW6 only - this needs clarification and rephrasing of this sentence.

Lines 259-261, lines 320-324: This work is not the first to use a multi-trait model in poultry selection (for example, the papers of Wolc et. all) - text needs to be rewritten.

Lines 295-296: There are no phenotypic correlations in Table 2 - please correct.

The places for corrections are marked in the manuscript.

Reviewer 4 Report

Dear authors,

This study analyses the genetic parameters related to body weights and egg production traits in Thai native crossbred chickens. The manuscript is well written and structured, the introduction provides sufficient background, the research design is appropriate, the methods are adequately described, the results are clearly presented, and the conclusions are supported by these results. For these reasons, I congratulate the authors for the study and the manuscript, and I consider that it can be published in its current form.

Author Response

Thank you very much for your support.

Reviewer 5 Report

It was seemed that the structure of the manuscript was poor at this time.

L16: I could not understand why “a multi-trait animal model” was a new finding from this study?

Materials and Methods: One Table  must be added showing the descriptive statistics (No. of records, mean, SD, and so on) for traits studied.

L118-124: Please describe the pedigree structure in this population. Was this population already selected?

L126-140: Please explain the meanings of the term “hatch” in this paper. How the authors control the effect of common environments, such as cage and maternal effects?

L130: E-MREML → EM-REML or EMREML. Furthermore, why the algorithm was used? How about using the average-information (AI) algorithm? How about the convergence?

L133-136: The expressions must be re-checked and then improved or modified.

L143-155 and L248: The expressing for calculation genetic progress the authors used was the expected selection response using a candidate’s phenotypic value for a given trait as the selection index, because the selection accuracy seemed to equal to the square root of trait heritability (h). Please denote the value for selection intensity. What did the term “gen” mean in this equation, because the equation seems not to express the expected “cumulative” genetic progress? Please explain the reason for setting the weight on the EBVs to be 0.25 for BW6, 0.25 for BrG6, 0.30 for NE240, and -0.20 for AFE. Please discuss about the accuracy of EBV using a multi-trait animal model constructed in this paper.

Table2: Values for phenotypic correlations should be added.

Figure 4: The authors must explain about the method of the significance testing used in detail.

Round 2

Reviewer 3 Report

Manuscript ID: animals-1488925

Title: Genetic Evaluation of Body Weights and Egg Production Traits using a Multi-Trait Animal Model and Selection Index in Thai Native Crossbred Chickens (Kaimook e-san2)

Recommendation: Minor Revisions

Title: since the authors use " Thai native synthetic chickens " in this manuscript the title should also be corrected to: " Genetic Evaluation of Body Weights and Egg Production Traits using a Multi-Trait Animal Model and Selection Index in Thai  Native Synthetic Chickens (Kaimook e-san2)"

Lines 18-20:  "Moreover, the method of estimating breeding value alone is not the way to genetic progress. The way can be selection on the basis of breeding value which is estimated by this model." - these sentences are unnecessary.

Lines 158-160: Which post - hoc test was used in GLM procedure - I renew my request for an addition in the text.

Author Response

To Reviewer,

We have revised our manuscript following your suggestion. Please review the revised MS and the Response to Reviewer files.

Sincerely yours

Wuttigrai Boonkum

Reviewer 5 Report

Point 5 L130: E-MREML → EM-REML or EMREML. Furthermore, why the algorithm was used? How about using the average-information (AI) algorithm? How about the convergence?

Response 5:

  • We have modified it to EM-REML in the revised MS.
  • EM-REML is the good approach to estimate genetic variances and parameters because it is an effective and general approach and is most commonly used for density estimation with missing data, like the Gaussian Mixture Model (multi-trait animal model).
  • About AI-REML, this approach performs as good in analysis as EM-REML, except that AI-REML takes more time to analyze than EM-REML because AI-REML need the first and second derivatives of the likelihood function, on the other hand, EM-REML need first derivative of the likelihood function only.
  • For convergence value, we set the convergence between Expectation step and Maximization step no more different than 10-10

> I concerned about the possibility the authors obtained the local solutions, but not global solutions, because the authors used EM algorithm. Therefore, I propose using AI algorithm and then comparing the results.

Point 7 L143-155 and L248: The expressing for calculation genetic progress the authors used was the expected selection response using a candidate’s phenotypic value for a given trait as the selection index, because the selection accuracy seemed to equal to the square root of trait heritability (h). Please denote the value for selection intensity. What did the term “gen” mean in this equation, because the equation seems not to express the expected “cumulative” genetic progress? Please explain the reason for setting the weight on the EBVs to be 0.25 for BW6, 0.25 for BrG6, 0.30 for NE240, and -0.20 for AFE. Please discuss about the accuracy of EBV using a multi-trait animal model constructed in this paper.

Response 7: the value for selection intensity is 20% per generation, gen in this study stand for year of birth of chicken, we explain the reason for setting the weight on the EBVs in the revised MS. Please see in lines 192-199. However, the accuracy of EBV is not examined in our study. This might be examined in the future.

> Accuracy of EBV, and therefore, accuracy of selection index is very important factor to predict the results of selection. The authors should discuss about this point in detail.

Point 9 Figure 4: The authors must explain about the method of the significance testing used in detail.

Response 9: The generalized linear model for unbalanced analysis of variance (PROC GLM) using the SAS package was used to compare EBV by generation. We indicated at line 186-187.  

> Did the author truly use “generalized” linear model? What distribution other than the Gaussian was assumed?

Author Response

To Reviewer,

We have revised our manuscript following your questions and suggestions. Please review the revised MS and the Response to Reviewer files.

Sincerely yours

Wuttigrai Boonkum

Round 3

Reviewer 5 Report

The manuscript has been improved.

Author Response

To reviewer,

Thank you very much for your comments and suggestions

Sincerely yours,

Wuttigrai Boonkum